# CLEVA-COMPASS: A CONTINUAL LEARNING EVALUATION ASSESSMENT COMPASS TO PROMOTE RESEARCH TRANSPARENCY AND COMPARABILITY

**Martin Mundt[1], Steven Lang[1], Quentin Delfosse[1] & Kristian Kersting[1,2,3]**
[1] AI and Machine Learning Group, Dept. of Computer Science, TU Darmstadt, Germany
[2] Centre for Cognitive Science, TU Darmstadt, Germany
[3] Hessian Center for AI (hessian.AI), Darmstadt, Germany
`{martin.mundt,steven.lang,quentin.delfosse,kersting}@cs.tu-darmstadt.de`

## ABSTRACT

What is the state of the art in continual machine learning? Although a natural question for predominant static benchmarks, the notion to train systems in a life-long manner entails a plethora of additional challenges with respect to set-up and evaluation. The latter have recently sparked a growing amount of critiques on prominent algorithm-centric perspectives and evaluation protocols being too narrow, resulting in several attempts at constructing guidelines in favor of specific desiderata or arguing against the validity of prevalent assumptions. In this work, we depart from this mindset and argue that the goal of a precise formulation of desiderata is an ill-posed one, as diverse applications may always warrant distinct scenarios. Instead, we introduce the Continual Learning EValuation Assessment Compass: the CLEVA-Compass. The compass provides the visual means to both identify how approaches are practically reported and how works can simultaneously be contextualized in the broader literature landscape. In addition to promoting compact specification in the spirit of recent replication trends, it thus provides an intuitive chart to understand the priorities of individual systems, where they resemble each other, and what elements are missing towards a fair comparison.

## 1 INTRODUCTION

Despite the indisputable successes of machine learning, recent concerns have surfaced over the field heading towards a potential reproducibility crisis (Henderson et al., 2018), as previously identified in other scientific disciplines (Baker, 2016). Although replication may largely be assured through modern software tools, moving beyond pure replication towards reproducibility with a factual interpretation of results is accompanied by tremendous remaining challenges. Specifically for machine learning, recent reproducibility initiatives (Pineau et al., 2021) nicely summarize how differences in used data, miss- or under-specification of training and evaluation metrics, along with frequent over-claims of conclusions beyond gathered empirical evidence impose persisting obstacles in our current literature. Similar conclusions have been reached in related works focused on specifics of reinforcement learning (Li & Talwalkar, 2019), neural architecture search (Lindauer & Hutter, 2020), human-centered machine learning model cards (Mitchell et al., 2019), or general dataset sheet specifications (Bender & Friedman, 2018; Gebru et al., 2018), which all make valuable propositions to overcome existing gaps through the creation of standardized best-practice (check-)lists.

It should thus come as no surprise that the emerging work in continual learning is no stranger to the above challenges. Superficially, continual learning has the intuitive objective of accumulating information and learning concepts over time, typically without the ability to revisit previous experiences, frequently also referred to as lifelong learning (Chen & Liu, 2018). However, there is no unique agreed-upon formal definition beyond the idea to continuously observe data, where the time component holds some practical implication on changes in the objective, the evolution of concept labels, or general statistical shifts in the data distribution. The majority of modern surveys ambiguously conflate these factors as a sequence of tasks (Parisi et al., 2019; Lesort et al., 2019; Hadsell et al., 2020; Lesort et al., 2020; Biesialska et al., 2021). Much in contrast to prevalent static benchmarks,

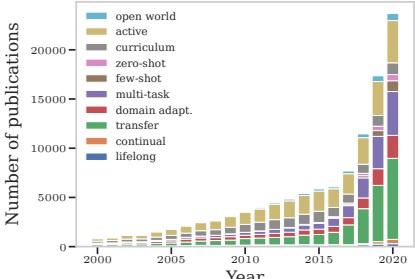 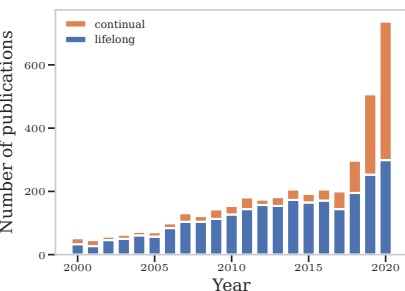

Figure 1: Per year machine learning publications. Left: cumulative amount across keywords with continuous components that influence continual learning practice, see Section 2. Right: increasing use of "continual", demonstrating a shift from the preceding emphasis on "lifelong". Data queried using Microsoft Academic Graph (Sinha et al., 2015), based on keyword occurrence in the abstract.

the question of reproducibility, interpretation of results, and overall comparability now becomes an even more complex function of nuances in employed data, training and evaluation protocols.

The latter fact has sparked various attempts at consolidation or critique. On the one hand, several works have made important suggestions for continual learning desiderata with respect to evaluation protocols (Farquhar & Gal, 2018; Díaz-Rodríguez et al., 2018; Kemker et al., 2018) and categorization of incremental training set-ups (van de Ven & Tolias, 2019; Lesort et al., 2021). On the other hand, empirical assessments have demonstrated that performance and comparability break down rapidly if often unexposed protocol aspects deviate (Pfülb & Gepperth, 2019; Delange et al., 2021). Following the broader exposition of the recent reviews of Mundt et al. (2020a) and Delange et al. (2021), evaluation becomes convoluted because intricate combinations of elements originating from various related machine learning paradigms affect continual learning practice. As the number of publications across these paradigms increases, see Figure 1, reproducibility, comparability, and interpretation of results thus also become increasingly difficult.

In this work, we follow in the footsteps of previous reproducibility works to promote transparency and comparability of reported results for the non-trivial continual learning case. Rather than adding to the ongoing discussions on desiderata or violation of assumptions, we posit that the development of distinct applications warrants the existence of numerous continual scenarios. Based on respectively highlighted evaluation nuances and their implications when absorbed into continual learning, we derive the Continual Learning EValuation Assessment (CLEVA) Compass. The CLEVA-Compass provides a compact visual representation with a unique two-fold function: 1. it presents an intuitive chart to identify a work's priorities and context in the broader literature landscape, 2. it enables a direct way to determine how methods differ in terms of practically reported metrics, where they resemble each other, or what elements would be missing towards a fairer comparison.

In the remainder of the paper, we start by sketching the scope of continual learning in Section 2, first by outlining the differences when going from static benchmarks to continual learning, followed by an exposition of evaluation nuances emanating from related machine learning paradigms. In Section 3, we then proceed to introduce the CLEVA-Compass and illustrate its necessity and utility at the hand of several continual learning works. Before concluding, we summarize auxiliary best-practices proposed in related prior works and finally discuss limitations and unintended use of the CLEVA-Compass. To encourage general adoption, both for prospective authors to add methods and for application oriented practitioners to identify suitable methods, we supply various utilities: a template for direct inclusion into LaTeX, a Python script, and the CLEVA-Compass Graphical User Interface (GUI), together with a repository to aggregate methods' compasses, all detailed in Appendix C and publicly available at `https://github.com/ml-research/CLEVA-Compass`.

## 2 THE SCOPE AND CHALLENGES OF CONTINUAL LEARNING EVALUATION

As already argued, there is no unique agreed-upon formal definition of continual learning. One of the few common denominators across the continual machine learning literature is the understanding

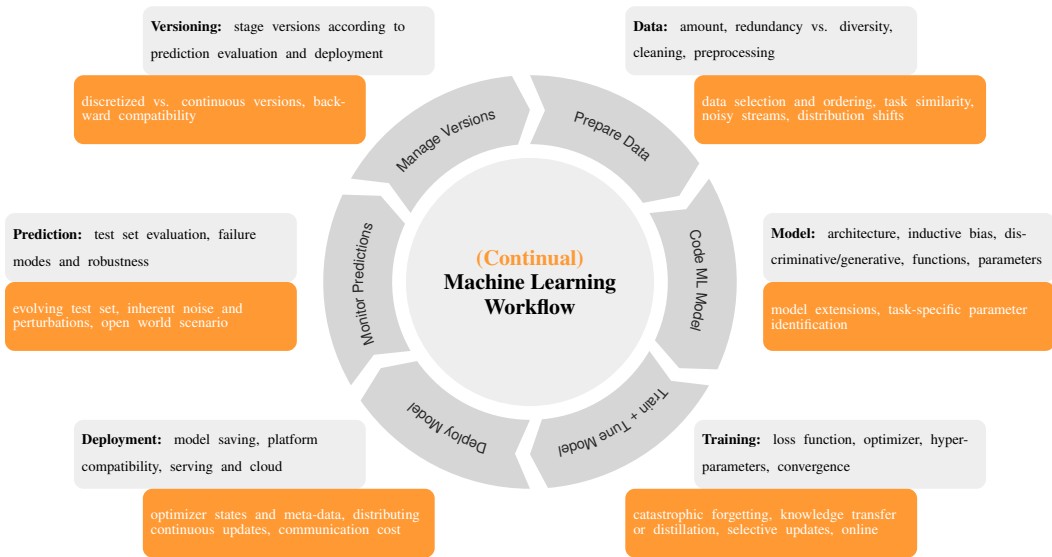

Figure 2: A typical (continual) machine learning workflow. The inner circle depicts the workflow advocated by Google Cloud (2021). On the outer circle we have added important, non-exhaustive, aspects to consider from the prevalent static benchmarking perspective (gray boxes) vs. additional criteria to be taken into account under dynamic evolution in a continual approach (orange boxes).

that catastrophic interference imposes a persisting threat to training models continually over time (McCloskey & Cohen, 1989; Ratcliff, 1990). That is, continuously employing stochastic optimization algorithms to train models on sequential (sub-)sets of varying data instances, without being able to constantly revisit older experiences, comes with the challenge of previously learned parameters being overwritten. As this undesired phenomenon is in stark contrast with accumulation of knowledge over time, preventing model forgetting is thus typically placed at the center of continual learning. Consequently, the focus has been placed on a model-centric perspective, e.g. by proposing algorithms to alleviate forgetting through rehearsal of data prototypes in training, employing various forms of generative replay, imposing parameter and loss penalties over time, isolating task-specific parameters, regularizing the entire model's functional or treating it from a perspective of dynamically adjustable capacity. We defer to review papers for taxonomies and details of algorithms (Parisi et al., 2019; Lesort et al., 2019; Hadsell et al., 2020; Lesort et al., 2020; Biesialska et al., 2021).

In practice, however, alleviating forgetting is but one of a myriad of challenges in real-world formulations of continual learning. Several orthogonal questions inevitably emerge, which either receive less attention across literature assessment or are frequently not made sufficiently explicit. A fair assessment and factual interpretation of results is rendered increasingly difficult. To provide the necessary background behind the statement, we briefly discuss newly arising questions when shifting from a static benchmark to a continual perspective and then proceed to contextualize conceivable evaluation protocol nuances in anticipation of our CLEVA-Compass.

## 2.1 FROM STATIC TO CONTINUAL MACHINE LEARNING WORKFLOW

To highlight the additional challenges in continual learning consider our visualization in Figure 2, depicting the benchmark inspired machine learning workflow as advocated by Google Cloud (2021). In the center, we find the six well-known sequential steps going from the preparation of data, to designing and tuning our ML model, down to the deployment of a model version to use for prediction. Naturally, these steps already contain various non-trivial questions, some of which we have highlighted in the surrounding gray boxes of the diagram. When considering popular benchmarks such as ImageNet (Deng et al., 2009), a considerable amount of effort has been made for each individual workflow step. For instance, assembling, cleaning and pre-processing the dataset has required substantial resources, a decade of work has been attributed to the design of models and their optimization algorithms, and plenty of solutions have been developed to facilitate efficient computation

or deployment. It is commonplace to treat these aspects in isolation in the literature. In other words, it is typical for approaches to be validated within train-val-test splits, where either a model-centric approach investigates optimizer variants and new model architectures, or alternatively, a data-centric approach analyzes how algorithms for data curation or selection can be improved for given models.

Much in contrast to any of the prevalent static benchmarks, establishing a similar benchmark-driven way of conducting research becomes genuinely difficult for continual learning. Instinctively, this is because already partially intertwined elements of the workflow now become inherently codependent and inseparable. Once more, we provide a non-exhaustive list of additional questions in the orange boxes in the diagram of Figure 2. Here, the boundaries between the steps are now blurred. To give a few examples: train and test sets evolve over time, we need to repeatedly determine what data to include next, an ongoing stream of data may be noisy or contain unknown elements, models might require new inductive biases and need to be extended, acquired knowledge needs to be protected but should also aid in future learning, and deployment and versions become continuous. In turn, setting a specific research focus on one of these aspects or attributing increased importance to only a portion allows for an abundance of conceivable, yet incomparable, implementations and investigations, even when the overall goal of continual learning is shared on the surface.

## 2.2 EVALUATION IN THE CONTEXT OF RELATED MACHINE LEARNING PARADIGMS

Although a full approach to continual learning should ideally include all of the underlying aspects illustrated in Figure 2, many of these factors have been subject to prior isolated treatment in the literature. We posit that these related machine learning paradigms have a fundamental and historically grown influence on present "continual learning" practice, as they are in themselves comprised of components that are continuous. More specifically, we believe that choices in continual learning can largely be mapped back onto various related paradigms from which continual scenarios have drawn inspiration: multi-task learning (Caruana, 1997), transfer learning and domain adaptation (Pan & Yang, 2010), few-shot learning (Fink, 2005; Fei-Fei et al., 2006), curriculum learning (Bengio et al., 2009), active learning (Settles, 2009), open world learning (Bendale & Boult, 2015), online learning (Heskes & Kappen, 1993; Bottou, 1999), federated learning (McMahan et al., 2017; Kairouz et al., 2021), and meta-learning (Thrun & Pratt, 1998). We capture the relationship with respect to set-up and evaluation between these related paradigms and continual learning in the diagram of Figure 3. For convenience, we have added quotes of the paradigm literature definitions in the figure. On the arrows of the diagram, we indicate the main evaluation difference and respectively how paradigms can be connected to each other.

As each paradigm comes with its own set of assumptions towards training and evaluation protocols, it becomes apparent why the quest for a strict set of desiderata can be considered as ill-posed for the continual learning hypernym. In the next section, we thus introduce the CLEVA-Compass, as an alternative that emphasizes transparency and comparability over strict desiderata. To further clarify this with an easy example, let us consider one of the most popular ways to construct protocols in the academic continual learning literature. Here, a continual investigation is set up by defining a sequence based on extracting splits of existing benchmark datasets and introducing them sequentially (Lesort et al., 2020; Biesialska et al., 2021; Delange et al., 2021). Such a scenario generally consist of individually introduced classes in image datasets like ImageNet (Deng et al., 2009), MNIST (LeCun et al., 1998), CIFAR (Krizhevsky, 2009), Core50 (Lomonaco & Maltoni, 2017), learning unique sounds (Gemmeke et al., 2017) or skills (Mandlekar et al., 2018; Fan et al., 2018) in sequence, or simply creating a chain across multiple datasets for natural language processing (McCann et al., 2018; Wang et al., 2019a;b) and games in reinforcement learning (Bellemare et al., 2013). Arguably, such a set-up is immediately derived from conventional transfer learning practice. Following the description of Pan & Yang (2010), the distinction between transfer and continual learning can essentially be brought down to the fact that both consider more than one task in sequence, but transfer learning focuses solely on leveraging prior knowledge to improve the new target task, typically a unique dataset or a set of classes, whereas continual learning generally intends to maximize performance on both prior and new tasks.

Even though these popular continual benchmarks already simplify the continuous data perspective to seemingly enable comparison akin to static benchmarks, there already persists an unfortunate amount of training, evaluation, and result interpretation ambiguity. For instance, Farquhar & Gal (2018); Pfülb & Gepperth (2019) argue that simply the knowledge of whether and when a new task

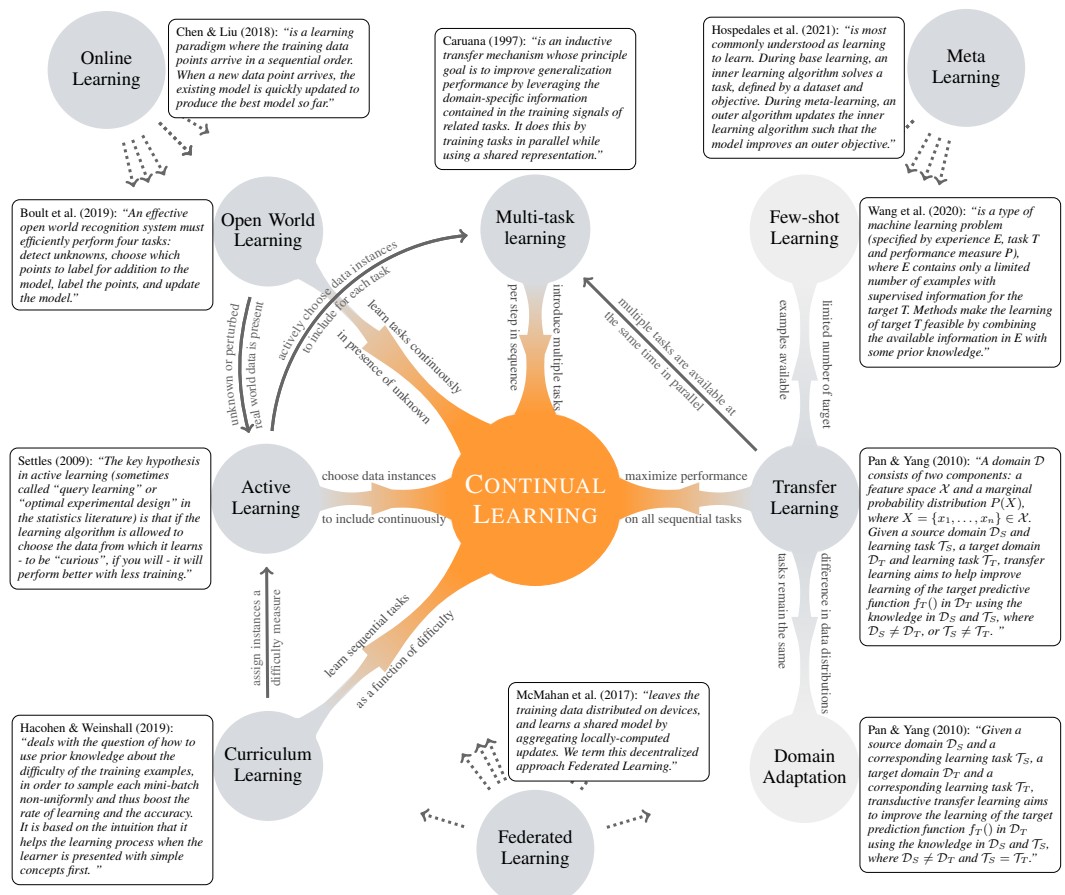

Figure 3: Relationships between related machine learning paradigms with continuous components. Popular literature definitions are provided in adjacent boxes. Arrows indicate the nuances when including perspectives into one another, showcasing the plethora of potential influences in construction of continual learning approaches. Outer nodes for federated, online and meta-learning can be combined and tied with all other paradigms. Note that only one arrow between nodes is drawn for visual clarity. Respectively highlighted distinctions can be constructed in mirrored directions.

is introduced already yields entirely different results, Mai et al. (2021) consider an arbitrary amount of revisits of older data and infinite time to converge on a subset as unrealistic, and Hendrycks & Dietterich (2019) contend that the presence of unknown noisy data can break the system altogether and should actively be discounted. Similar to the above inspiration from transfer learning, these works could now again be attributed to drawing inspiration from online and open world learning. In this context, online learning (Heskes & Kappen, 1993; Bottou, 1999) attempts to optimize performance under the assumption that each data element is only observed once, which can again be directly applied to "continual learning" (Mai et al., 2021). Likewise, open world learning (Bendale & Boult, 2015) allows for unknown data to be present in training and evaluation phases, which needs to be identified. Naturally, the latter can also be included seamlessly into continual learning (Mundt et al., 2020a). The list can be continued by iterating through all conceivable combinations of paradigms, illustrating the necessity for an intuitive way to make the intricate nuances transparent.

## 3 THE CLEVA-COMPASS

The subtleties of the relationships in Figure 3 provide a strong indication that it is perhaps less important to lay out desiderata for continual learning. However, we concur with prior works (Gebru et al., 2018; Mitchell et al., 2019; Pineau et al., 2021) that it is of utmost importance to have a clear

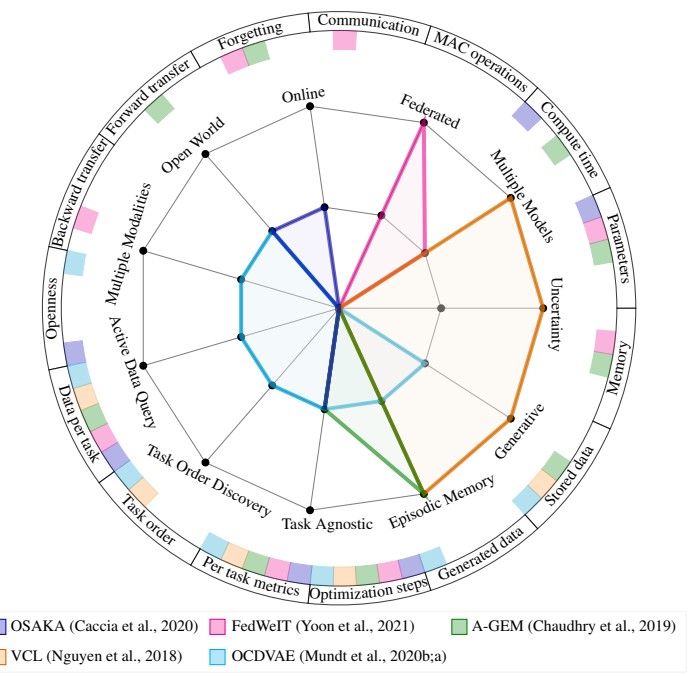

Figure 4: The Continual Learning EValuation Assessment (CLEVA) Compass. The inner star plot contextualizes the set-up and evaluation of continual approaches with respect to the influences of related paradigms, providing a visual indication for comparability. A mark on the star plots' inner circle suggests an element being addressed through supervision, whereas a mark on the outer star plots' circle displays an unsupervised perspective. The outer level of the CLEVA-Compass allows specifying which particular measures have been reported in practice, further promoting transparency in interpretation of results. To provide an intuitive illustration, five distinct methods have been used to fill the compass. Although these methods may initially tackle the same image datasets, it becomes clear that variations and nuances in set-up and evaluation render a direct comparison challenging.

exposition of choices and communicate research in a transparent manner. To maintain such a flexible research perspective, yet prioritize reproducibility and transparency at the same time, we introduce the Continual Learning EValuation Assessment Compass. The CLEVA-Compass draws inspiration from evaluations in prior literature across previously detailed related paradigms, in order to provide intuitive visual means to give readers a sense of comparability and positioning between continual learning works, while also showcasing a compact chart to promote result reproducibility.

The CLEVA-Compass itself is composed of two central elements. We first detail this composition, and then use it as a basis for a consecutive discussion on the compass' necessity and utility at the hand of five example continual learning methods. To provide a practical basis from the start, we have directly filled the exemplary compass visualization in Figure 4.

## 3.1 ELEMENTS OF THE CLEVA-COMPASS: INNER AND OUTER LEVELS

Figuratively speaking, the inner CLEVA-Compass level maps out the explored and uncharted territory in terms of the previously outlined influential paradigms, whereas the outer level provides a compact way to visually assess which set-up and evaluation details are practically reported.

**Inner level:** The inner compass level captures the workflow items of Figure 2 and reflects the influence of Figure 3 on continual learning. That is, for any given method, a visual indication can be made with respect to whether an approach considers, for instance, the world to be open and contain unknown instances, data to be queried actively, optimization to be conducted in a federated or online manner, as well as the set-up including multiple modalities or task boundaries being specified. Warranted by common practice, we further include an indication of whether an episodic memory is employed, a generative model is trained, and if multiple models are considered as a feasible solution.

In practice, we propose the use of a star diagram, where we mark the inner circle if a particular approach follows and is influenced by the relevant axis in a supervised way. Correspondingly, the outer circle of the star diagram is marked if the respective component is achieved in a more general form without supervision. It is important to note that this level of supervision is individual to each element. That is, just because a method has been used for an unsupervised task, does not automatically imply that all marked elements are similarly approached without supervision or vice versa. To give an example, one could consider continual unsupervised density estimation, where multiple models could each solve one in a sequence of unsupervised tasks/datasets, but a task classifier is required to index the appropriate model for prediction. We provide one respective example, including Figure 4's chosen methods outlined in the upcoming section, for all inner compass level elements to further clarify the role of supervision in Appendix B.

We have intentionally chosen the design of the inner compass level as a star diagram. These diagrams have been argued to provide an ideal visual representation when comparing plot elements, due to the fact that a drawn shape is enhanced by contours (Fuchs et al., 2014). Shape has been shown to allow a human perceiver to quickly predict more facts than other visual properties (Palmer, 1999). In addition, the closed contours of star plots have been discovered to result in faster visual search (Elder & Zucker, 1993), and in turn, supporting evidence has been found for resulting geometric region boundaries being prioritized in perception (Elder & Zucker, 1998). This is aligned with the primary goal of the inner compass level, i.e. exposing the conscious choices and attributed focus of specific continual learning works, in order for researchers to find a better contextualization and assess if and when comparison of methods may be sensible.

**Outer level:** Whereas the compass' star plot contextualizes the relation to linked paradigms with respect to comparability of set-up and evaluation protocols, the outer level places emphasis on the clarity with respect to reproducibility and transparency of specifically reported results. In essence, we propose to indicate a mark on the outer level for each measure that an approach practically reports in their empirical investigation. Together, the inner and outer levels thus encourage methods to be considered in their specific context, and comparisons to be conducted in a fair manner by promoting disclosure of the same practical assessment. Note that a practical report of additional measures beyond the required amount for a full overlap is naturally further welcome. We describe the list of measures on the outer CLEVA-Compass level in Table 1, where we also briefly summarize the relevance of each metric for continual learning and give credit to the corresponding prior works which have provided a previous highlight of their respective implications.

## 3.2 Necessity and utility: CLEVA-Compass in light of five examples

To provide the necessary intuition behind above descriptions for inner and outer CLEVA-Compass levels, we have filled the example illustration in Figure 4 based on five distinct continual learning methods: OSAKA (Caccia et al., 2020), FedWeIT (Yoon et al., 2021), A-GEM (Chaudhry et al., 2019), VCL (Nguyen et al., 2018), and OCDVAE (Mundt et al., 2020b;a). We visualize the compass for these continual vision methods because they have all initially conducted investigations on the same well-known type of image datasets, such as incrementally introducing classes over time in MNIST (LeCun et al., 1998) and CIFAR (Krizhevsky, 2009). Inspired by conventional machine learning practice, we might be tempted to benchmark and judge each approach's efficacy based on its ability to e.g. alleviate catastrophic forgetting. Our CLEVA-Compass directly illustrates why this would be more than insufficient, as each respective approach can be seen to differ drastically in involved set-up and evaluation protocols. For example, without delving into mathematical details, it suffices to know that FedWeIT prioritizes a trade-off with respect to communication, parameter and memory efficiency in a federated setting. In contrast, OSAKA considers a set-up that places both an emphasis on online updates as well as considering an open world evaluation protocol, where unknown class instances can be encountered with a given chance. Arguably, such nuances can easily be missed, especially the more approaches overlap. They are typically not apparent when only looking at a result table on catastrophic forgetting, but are required as context to assess the meaning of the actual quantitative results. In other words, the compass highlights the required subtleties, that may otherwise be challenging to extract from text descriptions, potentially be under-specified, and prevents readers to misinterpret results out of context in light of prominent result table display.

With above paragraph in mind, we note that a fully filled compass star plot does not necessarily indicate the "best" configuration. This can also best be understood based on the five concrete methods

Table 1: Description of conceivable measures on the outer level of the CLEVA-Compass. Further details and literature suggestions for mathematical definitions are summarized in Appendix A.

| Measure | Description and relevance for continual learning |
|---|---|
| Data per task | What data is introduced sequentially. The number of data instances is a primary indicator for sample efficiency and provides the context for e.g. few-shot settings (Fink, 2005). |
| Task order | The order in which tasks are introduced, even if randomly sampled in practice. The order has a significant impact on obtainable continual performance (Mundt et al., 2020a) depending on the constructed curriculum (Bengio et al., 2009). |
| Per task metrics | Task specific parts of reported losses or metrics allow for a deeper assessment of each task's evolution over time, e.g. *"new"* and *"base"* for first and most recent task, in addition to the overall average *"all"* (Kemker et al., 2018). |
| Optimization steps | The number of optimization steps is crucial to gauge empirical convergence. The number of optimization steps on revisited data also distinguishes sequences of continual offline and truly online scenarios (Mai et al., 2021). |
| Generated data | Amount of data that is generated, if any. The quality and number of data instances sampled from a generative model determine the effectiveness of rehearsal (Lesort et al., 2019) |
| Stored data | Amount of original data retained in a buffer, if any. Rehearsing instances becomes a trivial solution the more the buffer approximates the original dataset size (Delange et al., 2021). |
| Parameters | Amount of overall parameters. A trivial solution to continual learning would be to allocate increasing amounts of separate and independent parameters over time, motivating a desire for parameter efficiency (Díaz-Rodríguez et al., 2018). |
| Memory | How much memory is used. Provides a combined perspective on data storage and model parameter efficiency (Díaz-Rodríguez et al., 2018). |
| Compute time | Practically used computation time. Different algorithms and operations can consume dramatically different compute time, in additional dependence on hardware, even when implemented in the same software (Barham & Isard, 2019). |
| MAC operations | Number of multiply-accumulate operations are an alternative to reporting compute requirements, in a way that is not inherently tied to specific soft- and hardware (Sze et al., 2017). |
| Communication | Communication costs start to play a critical role in a distributed or decentralized federated perspective, where time spent on many rounds of communication can rapidly exceed that of model computations (McMahan et al., 2017). |
| Forgetting | The amount of forgetting is a way to quantify the difference between maximum knowledge gained about the task throughout the learning process in the past and the knowledge that is currently still held about it (Chaudhry et al., 2018). |
| Forward transfer | Forward transfer determines the influence that an observed task has on a future task (Lopez-Paz & Ranzato, 2017), quantifying the ability for "zero-shot" learning (Fei-Fei et al., 2006). |
| Backward transfer | Backward transfer (BWT) captures the improvement or deterioration an already observed task experiences when learning a new task (Lopez-Paz & Ranzato, 2017). |
| Openness | Openness of the world describes the proportion between data points that can be assumed to originate from the investigated data distribution and potentially unknown, corrupted or perturbed instances (Scheirer et al., 2013). |

in Figure 4. On the one hand, some aspects, e.g. the use of external episodic memory in A-GEM (Chaudhry et al., 2019) or existence of multiple models in VCL (Nguyen et al., 2018), could be interpreted as either an advantage or disadvantage, which is best left to decide for the respective research and its interpretation in scope of the underlying pursued applications. On the other hand, a discrepancy between methods in the CLEVA-Compass indicates that these continual learning approaches should be compared very cautiously and discussed rigorously when taken out of their context, and that much work would be required to make a practical alignment for these specific cases. As such, the CLEVA-Compass is not a short-cut to dubious "state-of-the-art" claims, but instead enforces transparent comparison and discussion of empirical set-ups and results across several factors of importance. In complete analogy, we remark that a filled outer level does not necessarily imply the most comprehensive way to report. Once more, this is because some practical measures may not apply to a specific method in a certain application context, e.g. reports of generated/stored data or communication costs when the method doesn't rehearse and is not trained in a federated setting.

## 4 UNINTENDED USE AND COMPLEMENTARY RELATED EFFORTS

We have proposed the CLEVA-Compass to promote transparent assessment, comparison, and interpretation of empirical set-ups and results. As the compass is a conceivable approach for a compact visual representation, there is a natural limit to what it should cover and what it should be used for.

One aspect that should be emphasized, is that the CLEVA-Compass is *not intended* to speculate whether a specific method *can in principle be extended*. Instead, it *prioritizes* which context *is taken into consideration in current practice* and lays open unconsidered dimensions. Specifically, if a paper does not report a particular compass direction, this may suggest that the method is not feasible in this scenario. However, it could also simply mean that the authors have not yet conducted the respective evaluation. A faithfully filled compass makes this transparent to researchers and should thus not be based on speculations. For instance, surveys (Parisi et al., 2019; Delange et al., 2021) commonly attribute several continual learning algorithms with the potential for being "task agnostic", but no corresponding theoretical or empirical evidence is provided. This has lead to empirical assessments, such as the ones by (Farquhar & Gal, 2018; Pfülb & Gepperth, 2019), to be "surprised" by the fact that certain methods do not work in practice when this factor is actually varied. Conclusively, if it is hypothesized that a method's capabilities extrapolate, it should be separated from the CLEVA-Compass' factual representation to avoid overly generalized, potentially false, conclusions. Its ultimate utility will thus depend on faithful use in the research community.

Apart from this unintended use of the CLEVA-Compass, there also exist several orthogonal aspects, which have received attention in prior related works. These works mainly encompass a *check-list for quantitative experiments* (Pineau et al., 2021), the construction of elaborate *dataset sheets* (Gebru et al., 2018), and the creation of *model cards* (Mitchell et al., 2019). These efforts should not be viewed in competition and present valuable efforts in the rigorous specification of complementary aspects. On the one hand, the check-list includes imperative questions to assure that theorems and quantitative experiments are presented correctly, think of the check-mark for reported standard deviations and random seeds. In Table 2 of Appendix D, we summarize why it is crucial to still consider these aspects, or why they are even particularly important in continual learning. On the other hand, dataset sheets and model cards present a verbose description of essential aspects to consider in the creation of datasets and models, with a particular focus on human-centric and ethical aspects. We stress that these perspectives remain indispensable, as novel datasets and their variants are regularly suggested in continual learning and the CLEVA-Compass does not disclose intended use with respect to human-centered application domains, ethical considerations, or their caveats. We give some examples for the latter statement in Appendix D. In addition to these central documents that capture primarily static complementary aspects of machine learning, the CLEVA-Compass thus sheds light onto fundamental additional features inherent to continual learning practice.

Due to their complementarity, we believe it is best to report both the prior works of the above paragraph *and* the CLEVA-Compass together. In fact, collapsing thorough descriptions of datasets and ethical considerations into one compact visual representation would likely oversimplify these complex topics. However, we acknowledge that the present form of the CLEVA-Compass nevertheless contains room to grow, as it is presently largely tailored to a statistical perspective of machine learning. Important emerging works that factor in recent progress from fields such as causality would thus be hard to capture appropriately. We imagine that future CLEVA-Compass updates may therefore include more elements on the inner star plot, or include additional reporting measures on the outer level, once corresponding advances have matured, see Appendix C for our future vision.

## 5 CONCLUSION

In this work, we have detailed the complexity involved in continual learning comparisons, based on the challenges arising from a continuous workflow and the various influences drawn from related machine learning paradigms. To provide intuitive means to contextualize works with respect to the literature, improve long-term reproducibility and foster fair comparisons, we have introduced the Continual Learning EValuation Assessment (CLEVA) Compass. We encourage future continual learning research to employ the visually compact CLEVA-Compass representation to promote transparent future discussion and interpretation of methods and their results.

## ACKNOWLEDGEMENTS

This work has been supported by the project "safeFBDC - Financial Big Data Cluster" (FKZ: 01MK21002K), funded by the German Federal Ministry for Economics Affairs and Energy as part of the GAIA-x initiative, the Hessian Ministry of Higher Education, Research, Science and the Arts (HMWK) project "The Third Wave of AI," and the German Federal Ministry of Education and Research and HMWK within their joint support of the National Research Center for Applied Cybersecurity ATHENE.

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

APPENDIX

We include four sections with additional information in the appendix. In Appendix A, we elaborate further on mathematical definitions for the measures on the outer CLEVA-Compass level, as described in Table 1 of the main body. In Appendix B, we provide a set of examples to further clarify the role of supervision in the various elements of the inner compass level. The five methods in the example main body compass illustration of Figure 4 are included in the discussion here, with additional examples serving the purpose to enhance the reader's intuition of how supervision could manifest in the remaining possibilities. Our CLEVA-Compass graphical user interface, code, and repository are then explained in Appendix C. Finally, as additional detail to the main body's Section 4, Appendix D provides further examples for, and underlines the indispensability of, complementary efforts with respect to the check-lists, dataset sheets and model cards introduced in prior related works.

## A   CLEVA-COMPASS METRIC: MATHEMATICAL DEFINITIONS

We have motivated the measures to report in continual learning in the description of the outer CLEVA-Compass level in Table 1 of the main body. There, we have also included references to prominent prior works that have outlined specific proposals. Although our work is not meant to serve as a full fledged review paper, we briefly quote corresponding mathematical definitions, in order to provide the reader with a more self-contained overview. For this purpose, we emphasize that the following provides *conceivable* ways to craft measures for specific applications, which is typically considered to be classification. In multiple of the below cases, several ways to report the same concept are thus listed. However, note that definitions are mostly either compatible and the exact expression transformable, or tailored to a particular use case.

We remark that we attempt to stay as close as possible to the authors' original notation (with only minor modifications to ensure consistency and resolve potential ambiguity in variable overlap), in order to facilitate understanding for the reader when going back for a more in-depth look at the original works. To make the explicit mathematical expressions accessible, we thus also follow the assumption that variable $t$ refers to observed tasks in $t = 1, \ldots, T$. This should not hinder generality of evaluation, as the knowledge of the task could be assumed to be known for testing purposes or strict boundaries between tasks could practically be further relaxed in below definitions.

**Per task metrics**

*Classification accuracy per task:* Lopez-Paz & Ranzato (2017) consider constructing an accuracy matrix $a \in \mathbb{R}^{T \times T}$, where $a_{i,j}$ is the test classification accuracy of the model on the j-th task after observing the last sample from task $i$. The final average accuracy is then defined as:

$$a_T = \frac{1}{T} \sum_{t=1}^{T} a_{T,t}\,. \tag{1}$$

*"Base", "new" & "all" classification accuracy:* Accuracy is split into three task specific parts. Closely following the description of Kemker et al. (2018): $\alpha_{new,t}$ is the test accuracy for task $t$ after task $t$ has been learned, $\alpha_{base,t}$ is the test accuracy on the first task (base set) after $t$ new tasks have been learned, $\alpha_{all,t}$ is the test accuracy of all the test data for the classes seen up to this point, and $\alpha_{ideal}$ is the offline accuracy on the base set, which is assumed to be the ideal performance. These accuracies can then be used to define the normalized quantities between $[0, 1]$:

$$\Omega_{base} = \frac{1}{T-1} \sum_{t=2}^{T} \frac{\alpha_{base,t}}{\alpha_{ideal}} \tag{2}$$

$$\Omega_{new} = \frac{1}{T-1} \sum_{t=2}^{T} \alpha_{new,t} \tag{3}$$

$$\Omega_{all} = \frac{1}{T-1} \sum_{t=2}^{T} \frac{\alpha_{all,t}}{\alpha_{ideal}}\,. \tag{4}$$

**Forgetting**

Forgetting quantifies the difference between the maximum knowledge gained about the task throughout the learning process in the past and the knowledge the model currently has about it. For classification, Chaudhry et al. (2018) define forgetting for the $j$-th task after the model has been trained up to task $t$ as:

$$f_j^t = \max_{i \in \{1, \cdots, t-1\}} a_{i,j} - a_{t,j}, \quad \forall j < t. \tag{5}$$

The average forgetting, after task $t$ has bee learned, is then defined as:

$$F_t = \frac{1}{t-1} \sum_{j=1}^{t-1} f_j^t. \tag{6}$$

**Forward transfer**

*Classification forward transfer:* The definition for per task accuracy has inspired expressions for forward and backward transfer (Lopez-Paz & Ranzato, 2017). For consistency with the above notation, we employ the notation according to Chaudhry et al. (2018). There, forward transfer (FWT) for classification is defined as the influence of the performance on future tasks $j > t$ when the model is learning task $t$:

$$\text{FWT}_{t,j} = a_{t-1,j} - \overline{b}_j, \tag{7}$$

where $\overline{b}_j$ is the accuracy on task $j$ of a random baseline. The average forward transfer is then defined as:

$$\text{FWT}_t = \frac{1}{t-1} \sum_{j=2}^{t-1} \text{FWT}_{j-1,j}. \tag{8}$$

Note, that $t$ is fixed to $j-1$ in the expression for the average.

*Learning Curve Area (LCA):* LCA is inspired by above definitions. Chaudhry et al. (2019) define the average $b$-shot performance after the model has been trained for all $T$ tasks as

$$Z_b = \frac{1}{T} \sum_{t=1}^{T} a_{t,b,t}, \tag{9}$$

where $b$ is the mini-batch number. LCA at $\beta$ is the area of the convergence curve $Z_b$ as a function of $b \in [0, \beta]$:

$$\text{LCA}_\beta = \frac{1}{\beta+1} \int_0^\beta Z_b \, db = \frac{1}{\beta+1} \sum_{b=0}^{\beta} Z_b. \tag{10}$$

That is, $\text{LCA}_0$ is the average 0-shot performance.

*Online codelength:* Motivated by the propositions of Kemker et al. (2018) and Chaudhry et al. (2019), a metric to measure the "adoption rate of an existing model to a new task" is introduced in the context of natural language processing (Biesialska et al., 2021). Termed the online codelength $\ell(\mathcal{D})$, a similar measure to area under the learning curve is given by:

$$\ell(\mathcal{D}) = \log_2 |y| - \sum_{i=2}^{N} \log_2 p\left(y_i \mid x_i; \theta_{\mathcal{D}_{i-1}}\right). \tag{11}$$

Here, $|y|$ is the number of possible class labels in the dataset $\mathcal{D}$ and $\theta_{\mathcal{D}_{i-1}}$ are the parameters trained on a particular subset of the dataset.

**Backward transfer**

Related to the concept of forward transfer for classification, backward transfer (BWT) describes the influence of the performance on previous tasks $j < t$ when the model is learning task $t$ (Lopez-Paz & Ranzato, 2017; Chaudhry et al., 2018):

$$\text{BWT}_{t,j} = a_{t,j} - a_{j,j} \,. \tag{12}$$

The average backward transfer can then be defined as:

$$\text{BWT}_t = \frac{1}{t-1} \sum_{j=1}^{t-1} \text{BWT}_{t,j} \,. \tag{13}$$

Note that a positive value for BWT implies a retrospective improvement of a task, whereas a negative value for BWT coincides with a quantification of model forgetting. Similar to the accuracy based definitions for forward and backward transfer, analogous expressions can be constructed on the basis of measuring task losses.

**Openness**

*Classification openness:* In the context of classification, Scheirer et al. (2013) provide a definition to capture the continuum between a completely closed world approach, where train and test examples are fully known to belong to the investigated task, and an open world, where various unknown instances can be observed. They propose to indicate the following quantity:

$$\mathbf{O} = 1 - \sqrt{\frac{2 \times N_{\text{train}}}{N_{\text{test}} + N_{\text{target}}}} \,. \tag{14}$$

This equation defines openness as a ratio between the number of classes used for training $N_{train}$ to the number of classes to be identified $N_{target}$ and the number of classes used in testing $N_{test}$.

*Openness as a probability to encounter unknown instances:* An alternative to the above definition for openness, particularly outside the context of classification, can be to indicate a probability with which out-of-distribution instances are encountered. For instance, Hendrycks & Dietterich (2019) define a probability for a sample to be perturbed or corrupted. Similarly, Caccia et al. (2020) indicate a sampling probability that dictates how many data instances are drawn with different, currently unknown, class labels.

**Other measures on the outer CLEVA-Compass level**

The majority of the other measures on the CLEVA-Compass' outer level are self-explanatory. For example, the size of the used dataset or number of optimization steps are simple scalar quantities to report. Nevertheless, there can exist alternatives to a straightforward report of e.g. number of model parameters or employed memory. For completeness, we mention some transformed alternative forms for these raw measures, according to the proposition of Díaz-Rodríguez et al. (2018).

- **Model size efficiency**: Instead of reporting the number of parameters of a model over time, it may also be sensible to provide a relative specification in terms of quantifying the growth of the model size:

$$\text{MS} = \min \left( 1, \frac{\sum_{t=1}^{T} \frac{\text{Mem}(\theta_1)}{\text{Mem}(\theta_t)}}{N} \right) \,, \tag{15}$$

  with $\text{Mem}(\theta_t)$ describing the number of parameters of the model at time step $t$.

- **Samples storage size efficiency**: Quantifying the memory used by the stored samples can similarly be reported in relation to the overall dataset size:

$$\text{SSS} = 1 - \min \left( 1, \frac{\sum_{t=1}^{T} \frac{\text{Mem}(M_t)}{\text{Mem}(\mathcal{D})}}{N} \right) \,. \tag{16}$$

  Here, $\text{Mem}(M_t)$ is the size of the stored samples in the memory, and $\text{Mem}(\mathcal{D})$ is the size of the overall observed dataset.

- **Computational efficiency**: As a combination of raw MAC operations and number of optimization steps, an additional quantity for computational efficiency can be expressed as the following ratio:

$$\text{CE} = \min\left(1, \frac{\sum_{t=1}^{T} \frac{\text{Ops}\uparrow\downarrow(\mathcal{D}_t)}{\text{Ops}(\mathcal{D}_t)}}{N}\right).$$

(17)

Here, $\text{Ops}(\mathcal{D}_t)$ is the number of operations needed to learn the (training) dataset $\mathcal{D}_t$, and $\text{Ops}\uparrow\downarrow(\mathcal{D}_t)$ is the number of operations required to do one forward and one backward pass on $\mathcal{D}_t$.

## B CATEGORIZATION OF SHOWN METHODS AND FURTHER EXAMPLES FOR THE CLEVA-COMPASS' INNER LEVEL

In the main body, Section 3 Figure 4, we have provided an example CLEVA-Compass illustration based on five continual learning methods. These methods have been deliberately picked to emphasize the compass' utility and necessity in light of the large exhibited differences. Correspondingly, the inner and outer levels have been described in detail in the main body, based on the related paradigms of Figure 3 and the detailed measures of Table 1. The example illustration highlights why the CLEVA-Compass is an important asset in identifying how methods relate in practice, exposing some of the typically unmentioned assumptions on set-up and evaluation, without requiring a detailed mathematical understanding of proposed techniques. However, our chosen example methods naturally only span a subset of overall possibilities. This is particularly true if we recall that there is a distinction between an unsupervised and supervised approach for every individual element of the CLEVA-Compass' star plot, see the description in main body Section 3.1.

In this appendix section, we thus provide examples for imaginable scenarios for every element on the inner compass level, both to enhance our intuition for the already shown methods and to emphasize why the distinction of supervision is crucial in the remaining settings. In the main body, we have included one such motivating example, an unsupervised continual density estimator, which in independence of being an unsupervised method itself, can have flexible requirements for supervision with respect to e.g. training multiple models in the continual setting. We emphasize, that the primary goal of such examples is to provide further clarification and that they are not meant to be viewed as exclusive definitions. For this purpose, the below list contains examples with and without supervision for each compass element, going clockwise from the "six o'clock" position of the compass and picking up the already visualized methods where applicable. Note that a short version of these examples is also provided in small "hover-over" tooltips in our CLEVA-Compass Graphical User Interface, which will be detailed in the upcoming section, Appendix C.

**Task agnostic:** A method is said to be task agnostic if for prediction in a deployed model it does not require any additional information for which task the data instance originates from. A supervised manifestation for such information, in independence of whether the task itself is supervised or not, would be to explicitly include a time-step or task label into the learning process to condition the prediction. The visualized OSAKA and A-GEM methods are examples of this, where a "context" or respective "task-descriptor" variable is explicitly conditioned on and later inferred. A fully unsupervised variant, or fully task-agnostic approach, would be able to inherently provide a correct prediction for any data instance from any previously observed task without any such information. In stark contrast, no mark on either supervised or unsupervised portion of the task agnostic star plot element would indicate that an approach is not capable of solving the task assignment challenge at all, implying that a task oracle is required for prediction. The latter is a typical, sometimes unexposed, assumption in many continual learning works that focus on other challenges, such as VCL, but is often argued to be unrealistic.

**Task order discovery:** One way to evaluate continual learning methods is to investigate the various measures on the CLEVA-Compass in a fixed sequence of benchmark data, corresponding to no visual mark on this star plot element for the majority of our example methods. An alternative, inspired by curriculum learning, would be to let the approach decide which task is meaningful to learn next. On the example of a classifier, rather intuitively, if a method can choose an improved task

order, it does so in a supervised fashion if it e.g. makes use of prospective class labels to distinguish which class would be best to learn next. An example of this is the visualized OCDVAE method, which constructs a class specific meta-recognition model to assess a similarity measure. On the flip side of this example, if the model discovers an improved task order based only on e.g. divergences or distances in any constructed feature space that do not require labels to compute, the task order could be said to be unsupervised.

**Active data query:** A traditional fixed sequence benchmark setting has no active data query component, observable in the majority of the visualized compass examples. Similar to above task order discovery, but inspired from the perspective of active learning, an alternative could be to actively query data (in independence of whether data is available in a pool, a stream, etc.). This way, an approach would actively choose what data instances to include next into optimization. As such, a measure of utility for prospective inclusion of a data instance is generally constructed. Correspondingly, this utility measure can either require presence of supervision, as shown in the class-specific meta-recognition model of OCDVAE, or can be entirely unsupervised.

**Multiple modalities:** If an approach only learns on one modality, such as text or images, then no indication of multi-modality is marked in the CLEVA-Compass. This is the setting for four of our example compass methods, with exception of OCDVAE that handles transformed audio and visual data in a supervised fashion. If the constructed system is able to handle multiple sources, a distinction between whether the multi-modality aspect is unsupervised or supervised condenses to the difference between whether or not the system requires a label on which modality an instance originates from, e.g. to condition a specific computation for this modality.

**Open world:** In an open world, the additional challenge for a learner is to robustly identify unknown, sometimes corrupted or perturbed, and potentially meaningless data instances. As an example for no indication of this aspect on the CLEVA-Compass, the majority of blackbox deep learning methods do not inherently posses robust identification mechanisms, which is the case in three of our shown example methods. If a mechanism is however included to recognize instances that deviate from the data distribution observed during training, then it would be supervised if its conception requires a class label, say a classifier entropy or similar supervised quantity. OSAKA and OCDVAE are two examples for the latter, the first of which identifies anomalies with respect to a supervised loss and the second of which relies on identifying anomalies with respect to a statistical meta-recognition model based on class-means. A respective unsupervised example could be a difference in e.g. a reconstruction loss or a divergence measure with respect to arbitrary feature spaces.

**Online:** In an online setting, a learner faces the additional difficulty of not being allowed to revisit observed data instances. No mark on the respective compass star plot thus implies that data is revisited. In continual learning, this corresponds to the common setting where several tasks are learned in sequence, but within each task several "epochs" are trained to convergence, as practically conducted in all but one of the chosen example methods. When moving to the online setting, the increased stochasticity alongside with data drifts become an increasing challenge. Example methods to address this scenario, by making sure that a consistent model is trained, could then be supervised or unsupervised. For instance, one could regularize parameter deviations over time according to a supervised importance measure, assume a supervised representation pre-training step as in OSAKA, or rely on various unsupervised quantities, such as exponential moving averages.

**Federated:** As one conceivable distinction, and once more independently of a task to be solved, federated scenarios could be supervised or unsupervised with respect to the federated implementation, akin to the examples of multiple models or modalities. For instance, we could assume that there exist a large number of devices in a network, but there are groups corresponding to different regions, different processing devices, labels on how many devices are participating actively, and so on. In that sense, communication in federated learning could provide additional supervised information that could be exploited to steer model training, optimize sub-model parts, or conversely go down a fully unsupervised route and simply average communicated updates without any label of the aforementioned characteristics. The example visualization if FedWeIT is an instance of the latter unsupervised setting.

**Multiple models:**   We have provided an example for the role of supervision in use of multiple models in the main body. We repeat this example here: multiple employed models could each solve a supervised or unsupervised task. In the supervision on the level of multiple models, an additional mechanism could be required to index the appropriate model for prediction, such as proposed in our FedWeIT example that employs specific attention based masks. If the correct model is automatically queried to provide correct predictions independently of which task an input belongs to this level is unsupervised. Finally, no visual mark on the compass corresponds to the scenario where only a single model is used.

**Uncertainty:**   Perhaps the least intuitive of the listed items, uncertainty could be believed to represent an inherently unsupervised quantity. We note however that uncertainty, especially in the way it is used in deep neural network approximations, is not always practically accurate and useful. As such, some uncertainty measures can typically require a need for calibration in order to provide meaningful values, e.g. the entropy of classifier predictions (note that we do not delve into a dispute of whether such a quantity should indeed be seen as a formal "uncertainty", but simply acknowledge the occurrence of such uses in the literature). Such calibration procedures can be interpreted as providing a supervised signal of what uncertainty "should look like". In contrast, if the method provides inherent uncertainties, such as argued in e.g. the fully Bayesian neural network viewpoint of VCL, it can be said to be unsupervised. Naturally, many approaches also do not provide any uncertainty estimates at all.

**Generative:**   Many methods that solve a specific task are not generative, e.g. consider a typical deep discriminative classifier of the form $p(y|x)$, where $x$ denotes data and $y$ denotes labels. A supervised generative variant would correspond to a model that learns the joint distribution $p(x, y)$ instead, whereas unsupervised generative models will learn or approximate only $p(x)$. Even when our objective is the classification example, the first of these variants will now also base decisions on the underlying nature of the data distribution. In current practice it is likely that an unsupervised generative model also primarily performs an unsupervised task (although we note that disentangled representations might in principle and prospectively allow classification approaches without supervised training). Nevertheless, the other way around, the distinction is still important because not every unsupervised task necessarily requires a generative model. We have included the shown Bayesian approach of VCL as an unsupervised generative example. However, it is worth mentioning that the authors also investigate a supervised variant in their work. As supervision could always be included as an additional term into fully unsupervised perspectives, we have marked the method on the outer unsupervised ring.

**Episodic memory:**   An episodic memory is constructed to effectively rehearse so called exemplars or prototype data instances. The challenge here is to assure that the constructed memory data subset is representative of the observed data as a whole. Intuitively, if a method does not employ an auxiliary episodic memory, no mark is indicated on the CLEVA-Compass. If the construction mechanism for this episodic memory relies on labels in the data, e.g. by approximating a per class mean in OCDVAE, then it is supervised. A straightforward unsupervised example would be to fill the episodic memory by sampling random data instances, such as suggested by A-GEM, or by employing unsupervised algorithms such as k-means clustering, as practically proposed in VCL.

## C    CREATING A CLEVA-COMPASS: CODE, GRAPHICAL USER INTERFACE AND METHODS REPOSITORY

To make the CLEVA-Compass as accessible as possible and disseminate in a convenient way, we provide three options for practical use.

1. We provide a **LaTeX template** for the CLEVA-Compass, making use of the TikZ library to draw the compass within LaTeX. We envision that such a template makes it easy for other authors to include a compass into their future submission, where they can adapt the naming and values of the entries respectively.

2. We further provide a **Python script** to generate the CLEVA-Compass. In fact, because the use of drawing in LaTeX with TikZ may be unintuitive for some, we have written a

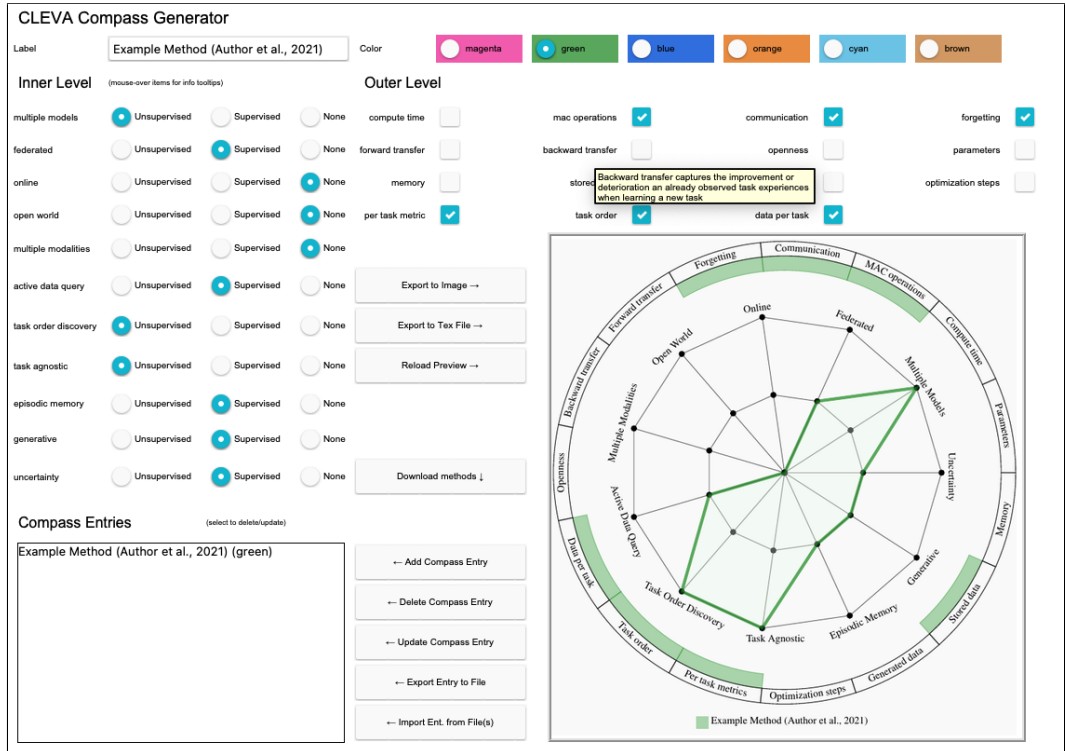

Figure 5: The CLEVA-Compass GUI. With the help of this application, users can interactively customize and construct their own CLEVA-Compass visualization or import existing ones.

Python script that automatically fills the above LaTeX template, so that it can later simply be included into a LaTeX document. The Python script takes a path to a JSON file that needs to be filled by the user with the CLEVA-Compass options. We further provide a default JSON file that is easy to adapt.

3. To further lower the barrier for dissemination and use, we also provide a **CLEVA-Compass Graphical User Interface (GUI)**. The GUI makes it easy for users to simply "click together" their desired compasses, save images or export to LaTeX, and conversely import already existing compass methods.

We refer to our public repository for the outlined code and the downloadable GUI: https://github.com/ml-research/CLEVA-Compass.

Code requirements and descriptions are explained in a corresponding README file there. In the following, we concentrate on the details of our CLEVA-Compass GUI. A visualization is depicted in Figure 5. We continue by explaining its main elements, functionality and its connection to the provided repository to accumulate methods.

**Creating CLEVA-Compass entries:** One of the key functionalities in the GUI is to quickly and intuitively add new methods (entries) into individual or existing CLEVA-Compass visualizations (see paragraph below for loading of existing methods). For this purpose, the GUI contains a *Label* field at the top left, intended to enter the method's name and authors, followed by the clickable options for all the items contained in the inner and outer compass levels. As detailed in the explanation of the main body Section 3.1, this involves the level of supervision on the inner level and the particular measures that have been reported on the outer level. Upon clicking the *Add Compass Entry* button, the method will be visualized and will show up on the bottom left list of visualized entries. The compass visualization will automatically adapt and scale with amount of entries (see also last paragraph for recommendations on amount of methods and colors). Should a user wish to adapt

or revise an entry, we offer buttons to *Delete Compass Entry*, *Update Compass Entry*, and *Reload Preview*. The remaining buttons provide various saving and loading functionality, detailed below.

**"Hover-over" tooltips for examples and hints:** We have detailed the levels and elements of the CLEVA-Compass throughout the main paper, with additional examples for further intuition. As it may be difficult to recall all the paradigm descriptions and described measures, we provide a "hover-over" functionality in the GUI. When placing the cursor above an item, e.g. a specific continual learning metric detailed in Table 1, a small box will show with the corresponding description. In the GUI Figure 5, this is illustrated at the example of backward transfer. In this way, a user is not require to swap back and forth between GUI and paper descriptions when designing their own CLEVA-Compass. In the same spirit, we have included small "example hints" for the inner level compass elements, in analogy to the descriptions of the ones provided in Appendix B.

**Saving and exporting functionality:** As described at the beginning of this section, there are multiple interface choices to make the compass accessible. In this spirit, our GUI also provides various exporting and saving functionalities. The most straightforward of these is the *Export to Image* button, which allows to save the visualized CLEVA-Compass in either `SVG` or `PNG` formats. In correspondence with our Python script, we also enable the option to *Export Entry to File*, which we build-upon to share and sync with our repository, detailed in the next paragraph. Lastly, to make it easy to include the Compass as TikZ code in a LaTeX document and thus allow for citation that is linked to reference lists, we finally provide the functionality to *Export to Tex file*. The latter creates the necessary TikZ code that generates the compass when LaTeX is compiled.

**Loading and the CLEVA-Compass repository to accumulate methods:** The final not yet described element of the GUI are the *Import Entry from File(s)* and *Download Methods* buttons. The *Import Entry from File(s)* functionality serves the purpose to enable users to load already existing CLEVA-Compass visualizations, in the form of loading their methods' JSON representations. As such, users will not have to replicate each and every method that has already been visualized in the CLEVA-Compass by hand. In addition to this, a list of existing methods, which at the point of writing this paper consists of the five methods of the main body, is provided in our public repository. By using the *Download Methods* button the GUI will automatically synchronize the up-to-date list of available methods and enable an interactive selection. Our vision is that prospective papers can contribute their own visualizations to this repository, so the amount of published methods and their CLEVA-Compass representations grows into a comprehensive repository. We strongly believe that this can help foster transparency in our community for prospective continual learning authors, but also in terms of creating a more straightforward overview of the set-up and evaluation practices of continual learning approaches for application engineers and practitioners. As a side note, we note that this attempt at cataloguing works and their "rolling" aggregation is separate from proposing prospective adaptation and extensions of the CLEVA-Compass (think of the example of including causality in our main body's outlook). For such major content and functionality updates, we subjectively envision a "discrete release" model, where prospective changes are encouraged to first undergo further stages of peer review, before being finally included into a CLEVA-Compass repository update. Although this may initially appear to slow down adoption of new methods, we argue in favor of this approach to limit the risk of a fixed set of researchers and a tiny portion of the community controlling such fundamental changes that steer the course of continual learning.

**A note on number of methods and colors:** Upon careful examination, one may note that we have chosen to include only six discrete colors as options for CLEVA-Compass visualizations. Rather than adding e.g. a color wheel to pick various colors, this presents a deliberate design choice. It is tied to the fact that we believe that there is little utility and value in visualizing more than six methods at maximum in one compass. The natural reason for this is that it becomes hard to distinguish methods after a certain point of overlap. However, we do not see this as a limitation, because star plots are well-known to provide excellent capabilities for visual comparison when placed side by side. In other words, when we place two CLEVA-Compasses side by side an easy and quick comparison is enabled, even if only one method were to be visualized in each.

Table 2: Auxiliary items from the machine learning reproducibility check-list (left-column) (Pineau et al., 2021) and their prevailing relevance for continual learning (right column).

| Paper reproducibility check-list: did you? | Additional importance for continual learning |
|---|---|
| *include the code, data, and instructions needed to* **reproduce** *the main experimental results* | Reproduction becomes even more challenging, as descriptions alone tend to overlook nuances in the precise way data is sampled continuously and models are adapted over time. |
| *specify all the* **training details** *(e.g. data splits, hyper-parameters, how they were chosen)* | Training details become particularly important, as precise data sequences determine comparability and methods can be subject to additional hyper-parameters that are now tunable per task. |
| *report* **error bars** *(e.g. with respect to the random seed after running experiments multiple times)* | In addition to the inherent optimization stochasticity, many continual methods introduce further random components, e.g. in random data sub set extraction or task order randomization. |
| *include the amount of* **compute** *and the type of* **resources** *used (e.g. type of GPUs, cluster, or cloud)* | Continual learning methods are often judged by their ability to adapt knowledge quickly or compute updates on-the-fly on data streams, which is directly tied to employed compute environments. |
| *use existing or curate/release new assets (e.g. code, data, models):* **cite** *the authors, mention the* **license***, include any* **new assets***, obtain* **consent** | Particular caution should be exercised as empirical continual investigations are frequently derived from available existing repositories and datasets. |

# D  IMPORTANCE OF EXISTING CHECK-LISTS AND DOCUMENTATION PROPOSALS FOR CONTINUAL LEARNING

We have introduced the CLEVA-Compass and have motivated its utility in the paper's main body. At the same time, in Section 4, we point out that there are several prior works that remain indispensable and can be considered orthogonal to our proposition. This is due to the fact that these works have both a different focus in terms of their intended use and have chosen a significantly different presentation format. Whereas our CLEVA-Compass provides a compact visual representation to contextualize continual approaches within related literature and contrasts their practical protocols, related works on *model cards* (Mitchell et al., 2019) and *dataset sheets* (Gebru et al., 2018) adopt a more verbose approach towards specification of elements surrounding data and intended model use. In addition to these efforts, the machine learning conference *reproducibility check-list* (Pineau et al., 2021) includes further general guidelines, that are meant to ground researchers in their practical machine learning reporting. The corresponding check-list items illustrate more general best-practice items when dealing with machine learning experiments. We provide a few examples in this Appendix Section to give additional explanations of why both the CLEVA-Compass and mentioned related works are complementary.

**Machine learning reproducibility check-list:** The items on the check-list have been proposed as guidelines towards general best-practice behavior in the context of recent reproducibility initiatives (Pineau et al., 2021). In particular, the main check-list's portion of empirical desiderata can be seen as at the center of commendable machine learning practice. This portion includes propositions for quantitative experiments, such as visualizing error bars, reporting random seeds, or detailing how hyper-parameters were tuned and chosen. An analogous argument can be made with respect to the inclusion of assumption explanations and proofs for involved theoretical statements. Consequently, the check-list can be regarded as auxiliary to the CLEVA-Compass, as the latter does not capture elements of general good practice with respect to figures, central tendency in results, or hyper-parameter tuning. In fact, we believe that many of the check-list items become even more important in the context of continual learning and the CLEVA-Compass. In Table 2, we thus indicate the entailed additional importance of selected check-list elements, when considered within continual learning.

**Model cards:** Mitchell et al. (2019) have suggested that every machine learning model should be accompanied with a so called model card. Such a model card is typically a one page overview, which summarizes: the choice of model, its intended use in terms of application, factors with respect to human-centric use, an indication of evaluated metrics, potential concrete results, along with any ethical considerations and caveats. In direct comparison with the CLEVA-Compass, the perspective of a model card could be summarized as primarily being tailored towards a static view of the human-centric application intent, in order to make deployment, limits, and ethical considerations more transparent and fair. The corresponding lengthy description provided in model cards is thus completely complementary to the continual factors captured in the CLEVA-Compass, which do not reflect a similar focus. In fact, we posit that it is even necessary to provide more detailed specification for ethical concerns, as they are hard to capture in a compact visual representation. To pick up concrete examples, model cards encourage the textual descriptions for: "person or organization developing the model", "model date, version and type", "license", "where to send questions", "intended use", human-centric "relevant factors including *groups, instrumentation and environments*" (i.e. personal characteristics, employed sensors, deployment environment), "unitary and intersectional results", "ethical considerations", "caveats", and so on. These factors are orthogonal to the CLEVA-Compass, but remain equally important to consider.

**Dataset sheets:** Gebru et al. (2018) have proposed to accompany every dataset with a dataset sheet. Such a document is naturally complementary to the CLEVA-Compass, as the latter merely captures the way in which data is used across continuous tasks, but not the underlying pipeline behind a dataset's creation and assembly. The prerequisite to specify the following aspects thus remains: composition, collection process, pre-processing, cleaning, labeling, distribution, maintenance, and ethical considerations. Dataset sheets are thus an important additional effort, which provide several questions and guidelines for a standardized way to document the various elements surrounding the former dataset aspects. Any new continual learning benchmark should therefore continue to provide a comprehensively documented dataset sheet.

**Software for replication:** Finally, for the sake of completeness, we also mention that there exist recent propositions for continual learning software tools, such as the Avalanche (Lomonaco et al., 2021) and Sequoia (Normandin et al., 2021) libraries on top of the popular machine learning framework PyTorch (Paszke et al., 2019). These efforts are naturally auxiliary in the sense that they do not expose a method's context, its caveats, ethical considerations or application intend, but rather just provide the means for pure replication of particular experiments (if data loading, methods, random seeds etc. are properly specified in the code).

