# OpenReview forum: "CLEVA-Compass: A Continual Learning Evaluation Assessment Compass to Promote Research Transparency and Comparability"
_ICLR.cc/2022/Conference — ICLR 2022 Poster_

### Official Review · Reviewer_JwDM · 2021-10-24

**Correctness:** 4
**Technical Novelty And Significance:** 3
**Empirical Novelty And Significance:** 3
**Recommendation:** 8
**Confidence:** 4

**Main Review:**

Strength:
- The paper is very well structured, clearly explaining the authors' concerns about the evaluation in Continual Learning. This structure helps to motivate the problem and present the solution adequately.
- I agree with the authors that comparing methods is not trivial. Different methods take different assumptions on the same problem, focusing on different metrics, which cause confusion and make the comparison tricky. CLEVA-Compass is a clever way to mitigate this problem by clearly identifying which metrics are reported and the contexts of the experiments.
- A significant problem raised by the authors is the transparency of the evaluation. For the most part, current incentives are to improve accuracy and decrease forgetfulness, often making unfair comparisons between methods or creating new metrics to "surpass" previous methods. Using CLEVA-Compass makes it more explicit about the context that each method is experimenting with, mitigating unfair comparisons.
- The CLEVA-Compass visualization is an excellent way to summarize information in a single visualization. Comparing more than two methods on the same inner level can be confusing, but multiple visualizations can solve this problem.

Concerns:
- My main concern with CLEVA-Compass is the selection of the different paradigms and metrics of the inner and outer levels. I understand that this is a vision of the authors, showing the areas and metrics relevant to Continual Learning. However, Continual Learning is inevitably evolving and merging with new scenarios, for example, self-supervision. As mentioned in the paper, CLEVA-Compass can evolve to adapt, but this can encourage people to create new factors (or metrics) that benefit their proposed method. Understanding that the latter is part of what you want to avoid with CLEVA-Compass: How are you planning to adapt your compass continuously? Or do you hope that the community will manage these modifications?
- A minor concern I have with CLEVA-Compass is the definition of the inner level points. I may not have understood it correctly, but I have two doubts:
    1. The definitions of each paradigm depend only on the definition given in Figure 3? These points may have ambiguous areas where concepts overlap and thus generate confusion.
    2. Regarding the 3 phases in the inner level, I think it can generate confusion when an unsupervised method is used in one area and not in another (as shown in the example of page 7, paragraph 1). If the area is ambiguous, in which part of the visualization the point is.
I am not an expert in all paradigms mentioned, so this ambiguity may not exist. Still, it is a concern that I have.


**Summary Of The Paper:**

In this work, the question arises of a better way to evaluate different methods in Continual Learning. The authors explain the problems of having a correct evaluation in Continual Learning, describing the complex scenario that Continual Learning confronts by being the intersection of multiple related areas. With this motivation, the authors present CLEVA-Compass or Continual Learning EValuation Assessment Compass, a visual representation that improves the comparison and transparency of different methods in Continual Learning. CLEVA-Compass includes two levels. The inner level shows which paradigms influence the method, and the outer level shows the setup and evaluation metrics presented in the method.

**Summary Of The Review:**

The paper addresses a crucial topic in Continual Learning, and I think it may interest many people. Evaluating and comparing different methods transparently is relevant not only for Continual Learning but for all areas of Machine Learning, as mentioned by the authors. My only concern is the questions I leave to the authors, which I hope they can answer.

---

> ### Author Response · Authors · 2021-11-12
> **The positive assessment and summary of strengths is appreciated. We respond to the remaining concerns and propose how to use the constructive feedback in our upcoming pdf revision and appendix extension.**
>
> We thank the reviewer for the assessment. We particularly appreciate the nice summary of strengths and are grateful for the pointers on where to provide further clarification.
>
> **Main concern: “selection of different paradigms and metrics of the inner and outer levels. Continual learning is inevitably evolving. How are you planning to adapt your compass continuously? E.g. avoid encouraging inclusion of factors that benefit a proposed method”**
>
> The raised questions highlight some interesting aspects with respect to the future. First, we agree that CL is inevitably evolving and continues to merge with more satellite formulations. In our current formulation, we have made an attempt at being as inclusive as possible, but naturally new dimensions will arise over time.
>
> We had considered the aspect of the evolution prior to submission. Initially, we had imagined that users could create requests to extend the compass with new metrics/inner level assumptions in our repository. However, saying this in humble honesty, it would put us in too powerful of a review position. We instead envision for new works that propose new dimensions to CL to first undergo conference/journal peer review, before warranting its definite inclusion into CLEVA. We acknowledge that this may not be ideal in terms of always being up-to-date with the latest cutting-edge. In other words, we are thinking of a “discrete release” model. Consequently, this will place some of our personal goodwill on the assumption that the reviewing process will successfully flag cases where authors suggest a dimension in a sole purpose to make themselves look good. We nevertheless feel it is preferable to us, or a static team, having too much influence on non-anonymously steering the direction of CL.  We understand that this may be a very subjective point and there likely exists no perfect solution. We will include our recommendation and rationale in our new appendix section, together with our new GUI and repository description (see response to reviewer RMSx).
>
> **Minor concern 1: “Definitions depend only on figure 3? There may be ambiguous areas where concepts overlap and thus generate confusion”**
>
> Yes, we have included definitions for orientation and to form the basis to understand the CLEVA-Compass. We acknowledge that the definitions contain some room for precise interpretation. This is a trade-off we had to deliberately take into account for the following two reasons:
>
> 1. Even though it may appear that there are some benefits in tailoring  definitions directly towards the continual learning context, we wanted to avoid re-defining all related paradigms (to avoid yet another layer of potential confusion). For this reason, we have thoroughly researched the literature for popular and heavily cited existing definitions and have then included the arrows between paradigms for relations.
> 2. Although there may be small room for ambiguity within the definition of one paradigm, we believe the general differences are defined sufficiently enough to not negatively impact the compass’ utility in terms of confusion on what concepts to mark. We have included the arrows in figure 3 to provide clarification on the crucial distinctions. The alternative would have been to select attempts at very precise definitions. However, we would argue that our paper’s described problem in CL arose from precisely such efforts towards strict and increasingly narrow definitions. These definitions would then potentially again disregard valid configurations, which would be against the spirit of our work.
>
> **Minor concern 2: “Phases of the inner level, confusion when unsupervised is used for one area, but not another. There may be ambiguity with respect to which part of the visualization the point is. Not an expert, so this ambiguity may not exist.**
>
> We understand the source of this concern. Initially, we had thought to not include the additional level of supervision in the compass to avoid potential confusion, but quickly noticed that this would be too restrictive and again render comparison unfair. When going through the examples, we noticed that the supervision is independent of the individual dimensions. In this process, we had then thought of whether there exist scenarios in which marking a respective point on the compass would clash with another or create large ambiguity. In our discussions, we did not come up with any major issues.  Naturally this does not necessarily exclude the general existence of edge-cases.
>
> Based on the question, we believe it will be valuable to expose this point to the reader, and by related request of reviewer BGUM, we will walk through some examples for each dimension (and our 5 examples) for this supervised/unsupervised part in detail in the appendix. We believe this should provide the necessary clarity. We have decided to also add “info boxes” into our upcoming GUI, where users will see the definitions for inner and outer levels, and example hints.

---

> > ### Comment · Reviewer_JwDM · 2021-11-22
> > **Appreciation to the response and the update of the document**
> >
> > I thank the authors for the detailed responses to my concerns and the answers to other reviewers' questions and feedback.
> >
> > I understand the reason for presenting the definitions the way the authors did. I think the authors have a good argument for leaving the definitions as it is. Exact definitions can generate more problems in the future since they could have flexibility problems.
> >
> > Regarding my second minor concern, I believe that both new sections (Appendix B and C) help to better understand each of the components and when it is a supervised/unsupervised/none, eliminating possible confusion in future applications.
> >
> > I appreciate the honesty of the authors regarding my main concern. Although the concern remains, I agree that it does not have an easy solution. However, it is good to know that the authors have this problem in their lists. The solution proposed in the paragraph *Loading and the CLEVA-Compass repository to accumulate methods* in Section C of the appendix seems adequate, where updates may be slow. Still, it is better to take small and safe steps than to make drastic changes without feedback.

---

### Official Review · Reviewer_BGUM · 2021-10-25

**Correctness:** 2
**Technical Novelty And Significance:** 3
**Empirical Novelty And Significance:** 3
**Recommendation:** 5
**Confidence:** 3

**Main Review:**

The work has nicely depicted different machine learning paradigms and their relationships with continual learning. Also it represents the difficulties and challenges arises from the continuous nature of this paradigm which makes the comparison of existing works challenging. The figures have efficiently summarize the main components in the paper. However, some main points of the research still remains unclear after reading the paper. The unclear parts are mentioned here:

1) In CLEVA there are two inner and outer circles which show whether a component is achieved in a supervised way or without supervision. Even-though an example is provided further explanation of this two layers can be beneficial to get the importance and difference of this two levels.

2) Highlighting influential paradigms and items that their extraction from text is challenging and sometimes misinterpreted is the only benefit of using CLEVA? The main positive effects and gains that CLEVA has still is not clear to me.

3) Authors stated that CLEVA can not be used for indicating the context that the proposed continual learning works can be applied, or showing superiority of one work versus other than, therefore how can it be used for assessment? In other word, what are the main advantages of CLEVA which can encourage researchers to use that for presenting their works.




**Summary Of The Paper:**

This work proposes CLEVA-compass which provides visual means to easily compare different continual learning works and provides a checklist to promote results reproducibility. Indeed, existing variations in the problem setup and evaluation of continual learning make the direct comparison of works in this field challenging. CLEVA tries to summarize each work in two different level: 1) each work's influential paradigms and items and 2) the measures reported in the work that can be used for reproducibly.

**Summary Of The Review:**

The main motivation and the benefits of the proposed compass is not clear. As an assessment tool the main advantages that this work can provide to the community is not clear to me. The better comparison of four works in terms of this compass in the text could be one way to show the superiority of the world of using CLEVA versus not using it.

---

> ### Author Response · Authors · 2021-11-12
> **The commendation on outlining the challenges and visualizations is appreciated. Based on the feedback on unclear parts we propose to include additional explanations in the appendix. Further clarifications are provided to the reviewer's questions.**
>
> We appreciate that  our exposition is valued and we are grateful for the feedback.
> Whereas the reviewer has commended our figures, we understand that the limited space in the main body led to some open questions that will help us in further improving our paper throughout the discussion period.
>
> **“Inner and outer circles, further explanation alongside the filled example” and “The better comparison of four works in terms of this compass in the text could be one way to show the superiority of the world of using CLEVA versus not using it”.**
>
> We agree that there will be additional value in providing a more detailed rundown of example methods. This will also give us the chance to further  highlight why the distinction between supervised and unsupervised on the inner compass star-plot level is important, beyond the brief existing paragraph in the main body. We will include more descriptions and summarize the details of the depicted methods in a new appendix section.
> We will also include info boxes in the upcoming GUI, see response to reviewer RMSx. Both should provide the readers/users with an even deeper understanding behind the choices and significance behind the compass.
>
> **“Benefit beyond avoiding misinterpretation”**
>
> We believe there are several benefits to the CLEVA-Compass:
>
> 1. It provides a concise visual overview of how methods compare, not only in terms of their set-up, but also how they have been evaluated in practice in particular works. This also means that researchers can easily show on what axes they may either completely differ, or go beyond previous works. To the best of our knowledge, there is no other existing work to quickly get such a direct overview.
> 2. Making the set-up assumptions on the inner level transparent is a key advantage. From the review, we get the impression that this is perceived as not valuable enough in itself. We would like to emphasize that the reason continual learning is in a dilemma is because these aspects are not typically exposed clearly. In fact, it is rather hard to think of all factors to report upfront, especially if one may think: “all my application requires is accuracy”. As the reviewer states, results being falsely interpreted and compared is one consequence. The conclusion based on the compass can be two-fold: either one can adapt the set-up to compare more fairly, or one can adapt the result discussion to take the differences explicitly into consideration, on the basis of the compass items. In either way, the benefit is a newly obtained transparency for the reader.  Analogous arguments can be made for the outer level of reported measures, with the small difference that usually reporting multiple metrics with maximum overlap is almost always a good idea.
> 3. From a different perspective, we would also like to emphasize that the CLEVA-Compass is not just intended for experts to include in future works. It also helps practitioners in identifying whether the assumptions/optimized metrics are suitable to their applications, rather than hoping things work in their scenarios and the unmentioned assumptions don’t result in disappointment. We will further gather visualizations of methods in CLEVA in our repository and encourage contributions when the compass is adopted in practice, which we will also detail in the appendix in the new GUI section.
>
> **“Authors stated that CLEVA can not be used for indicating the context that the proposed continual learning works can applied or showing superiority of one work versus another, therefore how can it be used for assessment?”**
>
> We believe there must have been a misunderstanding. We have naturally created the compass for the purpose of contextualizing works. That is, making transparent what the specific considered context is. In the paper, we mention several times that the compass highlights the subtle scenario assumptions and reported metrics, in order to prevent works being discussed and compared out of their respective context, and in a reverse argument promote comparisons that are fair and transparent. Our exact statement is that: “methods should be compared very cautiously out of their context”.
>
> We believe the source of confusion might have been (quote from the paper) “One aspect that should be emphasized in particular, is that the CLEVA-Compass is *not intended* to provide information for whether a specific method *can in principle be applied* in various contexts.” What we mean by this is that a reported method may only cover a certain part of the compass, but this does not mean that one could not also extend it to other parts of the compass in the future. It just means that original authors have not done this yet.  We actually see this as an additional benefit of the compass (in addition to above points). Researchers can use the compass to view where methods can be extended in the future. We will revise our wording to clarify.

---

> > ### Author Response · Authors · 2021-11-29
> > **A kind request for a short response**
> >
> > Dear reviewer, thank you once again for your original review and the constructive feedback.
> >
> > According to the ICLR timeline, it seems like the discussion period will be ending later today.
> > We understand that the discussion period in ICLR can be very time consuming. For this reason, we have done our best to stay concise, give directed responses limited to one box, and have followed up on concrete improvements with an uploaded revised pdf + supplementary.
> >
> > We believe to have addressed your raised concerns and questions, similarly to those of the other reviewers. We would greatly appreciate if you could provide a short response, even if it is just a brief note of acknowledgement. Any optional further feedback is naturally also welcome.
> >
> > Thank you in advance.

---

### Official Review · Reviewer_RMSx · 2021-11-01

**Correctness:** 3
**Technical Novelty And Significance:** 3
**Empirical Novelty And Significance:** 3
**Recommendation:** 8
**Confidence:** 4

**Main Review:**

The problem of relevant evaluation of continual learning approaches is a major one, and worth addressing. The authors do a great job presenting the different aspects of the evaluation (Fig 2) and explaining why they are meaningful. The distinction between static and continual evaluation is important, and definitely impacts the evaluation methods chosen and the metrics that can be used for comparing across them, as the authors rightly note.
I particularly liked this statement: "it is typical for approaches to be validated within train-val-test splits, where either a model-centric approach investigates optimizer variants and new model architectures, or alternatively, a data-centric approach analyzes how algorithms for data curation or selection can be improved for given models" -- it is a great way of comparing and contrasting different approaches. (I would argue that many of the arguments that the authors raise for continual learning apply for ML in general to different extents, actually, since in practice ML approaches are deployed in a way that is similar to continual learning -- i.e. production-ready ML systems are a often tacitly a form of continual learning without explicitly stating so.
I appreciated the open and transparent approach proposed for CLEVA. One thing I'm not 100% sure about is how practically it can be used, i.e. operationalized. This will undoubtedly become clearer when the code is released, but will it be via a UI or something similar? I feel like just showing the compass itself would not be enough for a full understanding of its scope, and how it can be applied.
I do find that there is a gap between theory and practice in the current proposal (which is highlighted quite well by the authors in section 4), and additional work is needed in order to better plan out its application in practice. i.e. "The CLEVA-Compass should thus not be used to conclude a method’s superiority and its utility will depend on faithful use in the research community." -- this is an important aspect of model evaluation that researchers will be looking for, and so if CLEVA doesn't allow it, this can get in the way of its widespread application.
This statement is not clear to me: "We believe it is best to avoid attempts at combining the prior works of the above paragraph with
the CLEVA-Compass." -- why is this the case?


**Summary Of The Paper:**

This paper proposes an approach for a more nuanced assessment of continual learning which provides a visual representation that enables the identification of a given method's context with regards to the broader literature, and enables the comparison of two methods in terms of reported metrics.

**Summary Of The Review:**

This is an interesting approach to evaluating continual learning. The transparent and multi-faceted approach is interesting and unique, and can be truly useful for those working in the community.
I feel that there is, however, a gap to be addressed between the more theoretical (and informative) usage of CLEVA and the more applied usage of it as a tool in guiding ML research. While section 4 of the paper does address this to some extent, the fact that the authors do not recommend using CLEVA to compare models, or alongside other works cited in that paragraph -- this would merit some additional thought and development.

---

> ### Author Response · Authors · 2021-11-12
> **The positive evaluation on uniqueness and relevance of our work is appreciated. We suggest further improvements based on the valuable feedback on GUI and distribution. Minor clarifications are provided**
>
> We thank the reviewer for the valuable feedback and are glad to see the reviewer agree on the relevance and uniqueness of our work.
>
> With respect to the concerns that the reviewer is not a 100% sure about, we are happy to provide additional clarifications below.
>
> **Code**
>
> We presently already include code in the uploaded supplementary. On the one hand, we provide a full LaTeX source (based on tikz drawing) template that can be directly copy pasted into a paper. As this may be unintuitive for some, we also provide a python script, that auto-generates the LaTeX representation by reading out a json file, where the user can add a list of items. For both options, we provide a joint readme file with instructions and 3 examples. As usability and distribution is very important to us, please also see the next point.
>
> **GUI & distribution**
>
> The reviewer’s valuable feedback has further inspired us to construct a graphical user interface, considering that we already have python code. The advantage of this is three-fold:
> 	1. Users can create compasses without delving into code.
> 	2. We have added small info boxes, when hovering over an item, to directly display descriptions (e.g. as contained in table 1 in the paper). and example hints.
> 	3. Our GUI will expose a synchronized list of methods available from our repository, once public. In addition to visualizing their own method, users can now interactively choose to include already existing works into their compass visualizations, rather than having to re-create them every time.
>
> Ideally, prospective works are encouraged to create requests to add more methods. We will add a new section in the appendix portion and re-upload the supplementary with the actual GUI before the discussion deadline.
>
> **Gap in proposed practice and usage “The CLEVA-Compass should thus not be used to conclude a method’s superiority”, the compass doesn’t allow it. “The authors do not recommend using CLEVA to compare models”.**
>
> We believe there is a misunderstanding. Our paper naturally proposes the CLEVA-Compass to compare works, that is its primary purpose, but in an effort to make these comparisons transparent and fair. We want to avoid apples to oranges comparisons.
>
> Where we raise awareness is that one should be cautious in drawing different lines in the compass and then claim a method is “SOTA” in “general continual learning”, which could be based on e.g. only a single reported metric. This ties back to our detailed explanation of section 2 and its figures. There are many formulations that can all be valid in different scenarios and many metrics of importance. In that sense, the compass is very important to avoid false SOTA claims, but promotes fair comparison with proper overlap, which presently causes a lot of confusion in continual learning (also in terms of reproducibility). To give an example, several works claim SOTA on “incremental classification”, but they do so by tweaking the set-up in using multiple models, compute, memory requirements, and evaluating differently, to the degree that they are no longer actually comparable to the related competitors, even on the same dataset.
>
>  The CLEVA-Compass aims to make all of this transparent, so that one can have an accurate view of how to fairly determine whether one work should be preferred over another.
>
> As an additional note, we believe the compass is not only useful for experts to actively fill in their algorithms for their next paper, but also for practitioners to start having a reference framework to intuitively see a “fit” of methods for their particular CL application purposes. Presently this is very challenging and tedious to figure out. If we just look at papers and see their results in a single table, we might be falsely driven to just pick whatever is “the best value”, which unfortunately in CL is very far away from the only important factor and the user is later disappointed because methods break down due to unmentioned parts.
>
> **“This statement is not clear to me: We believe it is best to avoid attempts at combining prior works of the above paragraph with the CLEVA-Compass — why is this the case? ”** and **”the authors do not recommend using CLEVA alongside other works cited in that paragraph”**
>
> It appears that the brevity of our statement may have lead to some confusion. In fact, we want to emphasize that users *should* use the CLEVA-Compass *and* document e.g. their dataset according to the cited prior works. This is because, as already detailed further in our appendix, these aspects are complementary and naturally go together. What we argue is that it would be very hard to capture e.g. all the nuanced dataset composition questions (examples: how it was created, curated, human involvement, ethics ...) also inside the CLEVA-Compass, which is why we do not believe a direct combination into *one visual representation* is immediately useful, but certainly a combination in the sense of *reporting both together* is.

---

> > ### Comment · Reviewer_RMSx · 2021-11-19
> > **Thank you for your responses**
> >
> > I appreciate the authors response, and believe that adding a GUI is a great contribution to making their tool more usable.
> > I also appreciate the clarifications that were made, and I think that adding those to the paper (to make certain statements clearer) would be helpful.
> > Thank you for the construction discussion!

---

### Official Review · Reviewer_FRys · 2021-11-01

**Correctness:** 3
**Technical Novelty And Significance:** 3
**Empirical Novelty And Significance:** Not applicable
**Recommendation:** 8
**Confidence:** 5

**Main Review:**

Pros:

The paper correctly identifies current challenges of defining CL research evaluations. The hypothesis that the abstract CL setting is a non-trivial multi-objective optimization problem has been well articulated in a review paper in Trends in Cognitive Sciences by Hadsell et al. 2020, but probably it has been given in several forms much earlier. The non-trivial term here has the technical meaning that no one solution exists which dominates all other solutions in terms of all objectives, hence trade-offs are needed. The only way to set trade-offs is to go to specific application domains and linearise the multiple objectives according to the needs of each domain. In that sense, the paper is correct that several rational definitions of CL are needed, assuming that the hypothesis outlined by Hadsell et al. is correct, as mounting evidence seems to suggest.

The paper coherently, albeit briefly, reviews the literature, ultimately arriving at a classification strategy. This type of review works help illuminate other sciences, e.g. neuroscience, and usually have dedicated publishing tracks in journals. Indeed, most practitioners in those disciplines relish review papers. Perhaps it is time for such tracks in machine learning main conferences, especially for subfields which deal with fundamental issues such as agreeing on definitions and evaluations.

Cons:

No formal system is given to precisely define terms, or at least outline how one should be developed. While this may be out of scope for one conference paper, the lack of such a system is an important source of confusion in CL research. At the end of the day any learning system should either optimise a precise criterion, at least approximately, or no criterion at all. This should be made clear upfront, with a discussion of all the assumptions such learning settings make.



Note to authors:

While the authors wish that the proposed categorization will not lead to a flurry of SOTA claims, it is safe to say that SOTA claims have already been made for most ways to cut the CL “cake”; even worse, SOTA claims are anecdotally thought to be a requirement for CL works to be published in main ML conferences. Hence, while the authors’ intentions are noble, the incentive structure may already be established against such hopes. At the end of the day, if quasi vacuous SOTA claims are what it takes for progress to be made in CL research, it is hard to argue against it, noble principles be damned. One may also take the view that SOTA claims made based on apple-oranges comparisons are not productive and may even stifle progress. However, the authors of this paper find a third option, proposing that such comparisons are productive exactly because they are correct if the “cake” is cut finely enough. While this entire paragraph of the review is somewhat in gest, it is included in order to serve as one of many possible subjective evaluations of the state of affairs in CL research, a path which was paved with good intentions.

Perhaps a fourth possible solution to this conundrum would be to acknowledge limitations of current algorithms and evaluation metrics, and naturally replace grand claims with precisely defined setups which aim to solve CL problems which other ML researchers encounter today:
For example:
Supervised or semi-supervised learning of very large-scale models on very large-scale (mostly unbalanced) and ad-hoc created datasets requires many passes through these datasets. Very little progress can be made in one pass, and SOTA continual learning algorithms do not seem to help.
Standard deep reinforcement learning *is* a continual learning problem with its own set of trade-offs due to agents being required to sample their own training data. Yet most current CL research does not seem to apply or help in practice.
Multi-agent RL is a highly complex data distribution shift problem. Training GANs or adversarially robust models against many types of attacks are natural dataset shift training scenarios, yet little help seems to be available from CL research.

The question arises: is a more structured classification of ongoing CL research into more fine-grained and fragmented  tracks going to improve the current state of affairs, making CL research more useful for ML in general? I personally find this question difficult to answer ahead of time, and will leave it for the research community.


Misc:
The A-GEM reference on page 8 (Chaudhry et al., 2018) may not be entirely accurate. I believe A-GEM was introduced in (Chaudhry et al., 2019), perhaps a small confusion could have occurred there.


**Summary Of The Paper:**

This paper argues that the goal of precisely formulating abstract desiderata for continual learning (CL) is ill-posed because different applications may always favour distinct scenarios. Following this logic, the paper proposes a categorisation system which aims to make CL research comparisons more structured. The paper goes on to review the challenges of meaningful evaluation of CL algorithms, particularly in terms of systematic comparisons of different works with somewhat different sets of assumptions, using the notion of “catastrophic forgetting” as a working example. Aspects which sometimes are overlooked in other machine learning research, e.g. dataset preprocessing and balancing, become important aspects in CL research and cannot be ignored. Furthermore, many machine learning paradigms may have natural extensions to non-i.i.d. learning scenarios, or at least various multi-task formulations of interest. Since many of these satellite formulations inherit metrics and assumptions of their respective paradigms, it naturally creates a problem formulation challenge for CL research.
The paper proposes a multi-objective radar/spider web chart (CLEVA-Compass) as a classification heuristic for CL research evaluation setups. The diagram is based on a well conceived collection of conceptual and practical desiderata for CL algorithms. The paper proposes how such a diagram could be used by future CL research and anticipates unintended uses. Limitations of the proposed classification strategy, as well as complementary challenges of dataset design, are discussed towards the end of the paper.


**Summary Of The Review:**

I believe it is timely to allow review papers into main ML conferences, so on balance I vote for acceptance. It is worth mentioning that several position papers cited in the text, e.g. Pineau et al. 2021, Mitches et al. 2019, Bender & Friedman, 2018 have all been published in journals and/or conference proceedings.

---

> ### Author Response · Authors · 2021-11-12
> **The detailed feedback, honest note and positive evaluation is appreciated**
>
> We thank the reviewer for the detailed feedback and positive evaluation. We appreciate the nice summary of our research, the honest additional note and the general recommendation for acceptance.
>
> Please see below comments for detailed responses.
>
> **Conference vs. special venue**
>
> We thank the reviewer in making a strong statement that it is timely to publish works akin to ours in conferences, similar to the mentioned works along related trains of thought. Given the generally rising interest in CL and various satellite formulations at these venues , we also believe that it is time to publish beyond novel algorithms.
>
> **(Con) no formal system is given to precisely define terms, or at least outline how one should be developed. While this may be out of scope for one conference paper, the lack of such a system is an important source of confusion in CL research.**
>
> The prospect of improving our understanding by developing more precisely defined terms is certainly intriguing. At the same time, as the lack thereof is mentioned as a “con” for our paper, we have to concur with the reviewer that an endeavor towards formal systems is out-of-scope of our conference paper. Especially, if not done extremely carefully and rigorously, one potential danger could again be the possibility of overly narrow definitions (and thus leading us back to the situation we are in now). In the current state, we believe it is already challenging to raise awareness and attempt to transparently lay out options and practical choices. Much of our effort has gone into making sure that our paper, besides its compact representation, provides a clear enough outline and proposition of the CLEVA-Compass. We see this as a first step to reduce the amount of confusion in CL research and genuinely hope that future works can draw inspiration from our approach to further build on it. In our view, this would eventually not only benefit CL research, but also the related satellite formulations in “broader” ML.
>
> **(Note) SOTA claims & assumptions should be discussed/made clear up front. The fourth possible solution could be to acknowledge limitations of current algorithms, evaluation metrics and naturally replace grand claims with precisely defined setups**
>
> We appreciate the transparent and honest remark of the reviewer. If our understanding is correct, the reviewer has added this paragraph without any major (negative) effect on the overall assessment. Nevertheless, the paragraphs contain very interesting aspects to discuss, which is why we return a response to the note.
>
> From our perspective, we had also thought about noble intentions and idealism involved in our proposition. In fact, drawing direct inspiration from the comprehensive and lengthy prior propositions on dataset sheets and model cards, we initially also thought about a similarly verbose way to elaborate on set-up and assumptions in detail. For now, we deviated from this initial take towards our present CLEVA-Compass form in an argument that is related to the points raised by the reviewer.
>
> First, providing textual descriptions on set-ups and assumptions apparently does not sufficiently happen right now in CL works, as grand SOTA claims seem to be anchored deeply into the incentive structure. We then asked ourselves if people would potentially adopt lengthy “continual learning sheets” into their works. As such, we believe that the barrier to use the CLEVA-Compass instead is practically lower (also in terms of consumed space). It further allows for intuitive visual comparison. It is this second point in particular that we believe warrants the existence of the CLEVA-Compass, as the CLEVA-Compass’ audience is not only intended to be the experts who conceive continual algorithms. We believe the compass will help practitioners to navigate and identify potential methods for their use-cases and applications. To keep the response concise here, please also see the response to reviewer 2 about us providing a GUI and collecting “already provided compasses for methods” in a common repository.
>
> **Citation Chaudhry 2019 vs 2018**
>
> The citation to A-GEM on page 8 is indeed incorrect, thank you for pointing this out. We had also cited Chaudhry 2018 on page 8 in table 1 for the introduction of the “forgetting” measure (introduced in the RWalk work).  This will be corrected.

---

> > ### Comment · Reviewer_FRys · 2021-11-25
> > **Thank you for addressing all points**
> >
> > It was very informative to get a glimpse of the development process for the compass and trade-offs involved, thank you for that!
> >
> > As my concerns have been well addressed, I am increasing my score to confidently recommend acceptance (8).

---

### Author Response · Authors · 2021-11-19
**Revised paper pdf and updated supplementary material has been uploaded**

Dear reviewers, once more we thank all of you for your detailed and constructive comments. Based on what we had already written in our responses, we have made use of the valuable suggestions to further improve our work.
With this comment, we note that we have uploaded an updated paper pdf and supplementary material.

A summary of the specific changes, improvements and extensions follows:

* **A-GEM citation fix**: we have fixed the A-GEM citation
* **Additional examples for the role of supervision on the inner compass level - the star plot: new appendix section B**: We have included a new 2.5 page long appendix section B to enhance the intuition for the inner compass level and its dimension of supervision. Here, we both further contextualize our already given compass visualization with its 5 chosen methods, and provide additional conceivable examples for each element on the inner level for every respective configuration of supervision.  We believe this will help readers to even better understand all the involved elements. We now also point to this section in the main body.
* **New CLEVA-Compass Graphical User Interface (GUI)**: we have developed a GUI to use with the CLEVA-Compass, in addition to our already provided LaTeX template and python script. We believe this will make the CLEVA-Compass even more widely accessible for prospective adoption. Apart from expected functionality to create, load, adapt, and  save CLEVA-Compasses, the GUI also contains hover-over “tooltip hint boxes”. These hints contain the explanations and examples of the main body and the new appendix section B, so that users can access and recall them in a convenient manner.
* **New appendix section C: CLEVA-Compass GUI details and public aggregation repository**: In addition to the supplementary upload, we have included another new 2.5 page appendix section C to both detail the GUI and discuss its tie to our public repository (which will be linked upon de-anonymization). In short, the public repository can serve the additional purpose of an aggregation platform for files of  already visualized CLEVA-Compasses. These will be automatically synchronized with the GUI, in order to provide an easy “import” functionality. Note that this is different from extensions to the compass, which we briefly also discuss here in accordance to what has already been stated in our review responses. We provide pointers to this new section in the main body.
* **Clarification on usage of CLEVA-Compass and related works, such as dataset sheets**: We have revised our formulation to more strongly emphasize that we encourage researchers to employ *both* the CLEVA-Compass *and* related works with respect to dataset sheets and ethical considerations. For instance, instead of saying: “We believe it is best to avoid attempts at combining the prior works of the above paragraph with the CLEVA-Compass”, we now state: “Due to their complementarity, we believe it is best to report both the prior works of the above paragraph *and* the CLEVA-Compass together.” (following with the necessary explanations in the respective paragraph). We believe this resolves any remaining sources for misunderstandings of our position with respect to related works.
* **Improvements towards clearer formulations with respect to comparison and (un-)intended use**:  Similar to above point, we understand that some of our formulations towards intended/unintended use could have been perceived in a misleading way. Without changing our message, we now provide more precise formulations. This primarily involves statements at the end of section 3.2 and in section 4. As one example, we have changed: “One aspect that should be emphasized in particular, is that the CLEVA-Compass is *not intended* to provide information for whether a specific method *can in principle be applied* in various contexts.” to the now easier to understand: “One aspect that should be emphasized, is that the CLEVA-Compass is *not intended* to speculate whether a specific method *can in principle be extended*. Instead, it *prioritizes* which context *is taken into consideration in current practice* and lays open unconsidered dimensions. “ Our newly revised formulations should help to avoid the potential raised sources of confusion with respect to the CLEVA-Compass not being indicative of the context (we emphasize that this is a primary advantage of the compass) or us discouraging its use for comparison (which is again one of the primary purposes of the compass to make comparison transparent, fair and enable respective discussion to be more factual.)

---

### Decision · Program_Chairs · 2022-01-20

**Decision:**

Accept (Poster)

**Comment:**

This review paper presents a way of comparative assessment of continual learning. Reviewers all agreed that this work is interesting, unique with comprehensive coverage of the CL space. The proposed categorization, CLEVA-Compass, and its GUI have great potential to facilitate future CL work.